# Home-Based Aerobic and Resistance Exercise Interventions in Cancer Patients and Survivors: A Systematic Review

**DOI:** 10.3390/cancers13081915

**Published:** 2021-04-15

**Authors:** Ladislav Batalik, Petr Winnige, Filip Dosbaba, Daniela Vlazna, Andrea Janikova

**Affiliations:** 1Department of Rehabilitation, University Hospital Brno, 62500 Brno, Czech Republic; winnige.petr@fnbrno.cz (P.W.); dosbaba.filip@fnbrno.cz (F.D.); vlazna.daniela@fnbrno.cz (D.V.); 2Department of Public Health, Faculty of Medicine, Masaryk University, 62500 Brno, Czech Republic; 3Faculty of Medicine, Masaryk University, 62500 Brno, Czech Republic; janikova.andrea@fnbrno.cz; 4Department of Neurology, University Hospital Brno, 62500 Brno, Czech Republic; 5Department of Internal Medicine–Hematology and Oncology, University Hospital Brno, 62500 Brno, Czech Republic

**Keywords:** cancer, rehabilitation, exercise, home-based exercise, cancer survivors, cardiorespiratory fitness, physical activity

## Abstract

**Simple Summary:**

Exercise interventions are increasingly being recognized as an important part of treatment and supportive care for cancer survivors. Although the beneficial effects of exercise interventions on a number of physical and psychosocial factors, which can mitigate the effects of cancer treatment, have been described, several barriers remain that affect the use of exercise interventions. An alternative form, home-based (HB) exercise, has the potential to overcome several accessibility barriers that limit cancer survivors from participating in exercise-based interventions under professional supervision. In addition, in the current situation associated with the COVID-19 pandemic, alternative remote access models and their variations are strongly supported. Since a comprehensive review of HB exercise interventions is lacking, we reviewed the current literature on the role of HB exercise during the rehabilitation period in a cancer survivor population. This article identifies the literature on the health effects of HB exercise interventions and evaluates studies’ methodological quality. The obtained results provide a starting point for further research directions and addressing challenges in this research area.

**Abstract:**

Cancer is a chronic disease requiring long-term treatment. Exercise interventions are increasingly being recognized as an important part of treatment and supportive cancer care for patients and survivors. Previous reviews have evaluated the benefits of exercise interventions in populations of patients under supervision at a center, but none have explored the possibilities of a home-based (HB) approach in exercise during cancer rehabilitation and the period immediately following the end of cancer treatment. The aim of this descriptive systematic review was to identify the literature focusing on the health effects of HB exercise interventions in cancer survivors and to evaluate the methodological quality of the examined studies. Relevant studies were identified by a systematic search of PubMed and the Web of Science until January 2021. Nine randomized controlled trials were included. Most studies were on aerobic and resistance exercises, and the frequency, duration, intensity, and modality varied across the different interventions. Improvements in cardiorespiratory fitness (CRF), physical activity (PA) levels, fatigue, health-related quality of life (HRQOL), and body composition have been reported. However, all the studies were limited in methodology and the reporting of results. Nevertheless, the evidence in this new area, despite the methodological limitations of the studies, suggests that HB exercise interventions are feasible, and may provide physiological and psychological benefits for cancer survivors during the rehabilitation period. A methodologically rigorous design for future research is essential for making progress in this field of study.

## 1. Introduction

The International Agency for Research on Cancer estimates that there were 19.3 million new cases of cancer and nearly 10.0 million cancer deaths worldwide in 2020 [1]. The growth trend in new cases is expected to continue, and in 2040, the global cancer burden will be 50% higher than in 2020. Efforts to build sustainable approaches to cancer treatment and prevention are crucial. Improving treatment and supportive cancer care is responsible for reducing mortality and increasing the life expectancy for cancer survivors [2]. However, for many cancer survivors, treatment has long-term negative physical and/or mental side effects, such as cancer fatigue, muscle atrophy, and cardiotoxicity [3,4,5,6]. Based on these effects, cancer is considered a chronic disease that requires long-term systematic management; so, evidence-based rehabilitation interventions tailored to this population are needed [7].

Exercise interventions are increasingly recognized as an important part of treatment and supportive care for patients and cancer survivors [8]. The scientific evidence from previous systematic reviews suggests that exercise interventions provide a number of physical and psychosocial benefits that can mitigate the effects of cancer treatment. The benefits have been shown to include the improved cardiorespiratory fitness (CRF), health-related quality of life (HRQOL) of physical and immune function, and reduced fatigue and depression [9,10,11]. Despite the recognized benefits, a number of barriers affect the implementation of exercise interventions in cancer survivors: legal and organizational barriers, including a significant lack of specialized rehabilitation services, low awareness, or lack of referral from health-care providers [12]; barriers in the area of patients include inconvenient place, time of day, or insufficient capacity to offer physical exercise rehabilitation programs [13].

An alternative form of exercise intervention, home-based (HB) exercise, for cancer survivors has the potential to overcome several accessibility barriers that limit patients from participating in traditional center-based (CB) exercise interventions under professional supervision. As is well known, recent studies of HB exercise interventions have provided evidence of feasibility and safety, supporting that HB exercise appears to be a suitable alternative to CB programs [14,15]. Particularly during the current COVID-19 pandemic, an alternative model of remote access and its variations are strongly recommended in chronic diseases [16].

A comprehensive overview of HB exercise interventions for cancer survivors is not currently available; therefore, we reviewed the current literature on the role of HB exercise interventions during the rehabilitation period. It is important to identify the direction and challenges gained from these studies for the preparation of future research in this area. The aim of this descriptive systematic review was to identify the literature on the health effects of HB exercise interventions in cancer survivors and to evaluate the methodological quality of the examined studies. This systematic review discusses the exercise interventions, participants, exercise adherence and compliance, adverse events, and study outcomes.

## 2. Materials and Methods

### 2.1. Search Strategy

Search strategies and article selection criteria were searched from January 2011 to January 2021 through the PubMed database and the Web of Science metasearch engine. L.B. and D.V. developed a search strategy. The selection process involved searching with the keywords as follows: (“Cancer Survivors”) OR {(Neoplasms OR Neoplasms OR Cancer) AND (Survivors OR Survivors OR Patient OR Patients OR Persons)} A (“Exercise Therapy” OR Exercise Therapy OR Physical Activity OR Exercise OR Training OR Physical Rehabilitation OR “Rehabilitation”) AND (home OR mHealth OR eHealth OR internet OR mobile OR intelligent OR tele OR remote OR cardio-oncological OR cardio-oncological OR oncological rehabilitation OR hybrid).

L.B. then conducted an initial search, and, in consultation with D.V., suitable articles were subsequently selected. The articles were further evaluated by two reviewers, P.W. and F.D., based on the abstracts. In the next step, the full-text detailed articles that met the criteria for inclusion were thoroughly revised. The inclusion criteria for the article were as follows: studies published in English, randomized control design of the study with active or usual care; at least eight weeks of intervention period; evaluation of the effectiveness of the intervention on CRF. In addition, a population of cancer patients or survivors who had recently completed treatment for any cancer (except adjuvant treatment) was identified. The interventions that met the criteria were: aerobic and resistance HB exercise programs that included results and analysis. Interventions that included only resistance exercises were excluded. The results that met the criteria were all effective interventions associated with health and subsequent treatment that was provided during the rehabilitation period.

The description of the HB exercise intervention consisted of the identification of the following components: (1) monitoring exercise and (2) the feedback method through which the data were collected and analyzed. Further, description of the exercise was included: prescription of intensity and time using wearable sensors (e.g., heart rate monitors, accelerometers, and pedometers). In another area of teleconsulting, the variability of controlling, educating, and motivating methods for physical exercise were monitored. How often the specialist in the hospital center obtained the exercise data was described.

Duplicate studies or studies that did not publish the full-text and/or description of the HB method of exercise intervention were excluded. Systematic reviews, meta-analyses, or expert comments were excluded from the review, but reference lists of localized articles on the topic were screened. The full texts of the selected studies were further evaluated and if there was a disagreement between the two reviewers P.W and F.D., lead author L.B. decided on the suitability of the article for inclusion in the review.

### 2.2. Data Extraction and Quality Assessment

The descriptive characteristics of each study were extracted, including the study design, recruitment specification, and participants’ characteristics. Descriptive data on the exercise program were also extracted, including the length of the program, the duration, frequency, and intensity of the exercise, adherence, and/or compliance to the exercise prescription. Finally, the results of the evaluation and the effect of the intervention on health outcomes were extracted. The methodological quality of the involved studies was evaluated using the TESTEX tool [17].

The TESTEX tool was selected due to its reliability and suitability for exercise researchers as it facilitates a comprehensive review of exercise training studies. The advantage of this tool is that it considers the criteria of blinding of participants or study investigators, which are extremely difficult to implement in exercise-based studies. TESTEX includes 12 criteria, some of which can be evaluated with more than 1 point, allowing a maximum of 15 points. A total of 5 points can be obtained for the methodological quality of the study, while 10 points can be obtained for study reporting. The higher the score, the better the quality of study and study reporting. The studies were classified according to the average score as: high quality as 12 points or more, good quality as 7 to 11 points, and low quality as 6 points or less. High-quality interventions were defined as highly relevant, reproducible, and very well methodically described, with an excellent report of results [17]. Good-quality interventions were defined as moderately clinically relevant, limitations in reporting results, and good reproducibility for further experiments. The low-quality interventions showed substantial limitations regarding relevance and a method used with low reproducibility. This individual approach was chosen because the validated TESTEX cut-off scales were not yet recommended.

Four key features of methodological assessment were qualified for all eligible studies: the use of randomization for group allocation, the use of an unbiased randomization method, blinding of assessors, and the use of intention-to-treat analysis.

The studies were also assessed with respect to the categories of primary and secondary outcomes used in HB exercise interventions. The effect of the HB exercise intervention based on the CRF assessment, completion rate, and safety assessment of the intervention, which focused on the occurrence of adverse events, was described. Adverse events were identified as mild or moderate events that could lead to further hospitalization. Mortality was defined as a serious adverse event and evaluations were included.

## 3. Results

We identified a total of 1510 records when searching the database and metasearch engine. Screening of titles and abstracts showed that 287 publications did not meet the entry criteria. The remaining 37 publications were subjected to a detailed examination of the entire text, from which 28 publications were excluded. Of the 28 excluded publications, six publications were excluded because they were performed before or during cancer treatment, and five were excluded as they described CB exercise rehabilitation or a hybrid approach combining HB and CB exercise. Another four publications reported insufficient description of HB exercise intervention, three publications reported less than eight weeks of intervention, three publications reported a single-group design, three publications lacked randomization, two publications were based on resistance exercise only, and two publications did not examine exercise effectiveness. A final nine publications were included in this systematic review based on the study inclusion criteria. An overview of the study flow process is shown in Figure 1.

### 3.1. Studies Included

The characteristics and findings of the nine included studies [18,19,20,21,22,23,24,25,26] are listed in Table 1. A description of the methodological quality of each study, completion rate, incidence of adverse events, and adherence rate with exercise intervention within the studies are provided in Table 2. Two of the nine studies were conducted in Canada [18,19] and in the Netherlands [20,24]; one each was conducted in the United Kingdom [21], Denmark [22], France [23], Norway [25], and the United States [16]. In the period from 2011 to 2015, three studies were published, and six studies were published in the period from 2016 to 2020.

### 3.2. Sample Size and Recruitment

The calculation of the sample size was reported in eight studies; in one of the nine studies, the calculation was not included [22]. The number of participants in the included studies ranged from 25 to 230 participants. The average age was 57 years. In total, there were 630 participants included in the systematic review. Several studies [19,20,26] pointed to recruitment difficulties and did not include the intended number of participants. For example, Pinto et al. [26] reported that “Despite using various recruitment methods we were unable to randomize the required sample size” in the colorectal cancer survivors population and discussed the subsequent uses of national cancer and tumor registries, which may be more effective in reaching participants in the future [26]. Gering et al. noted that recruitment was most limited due to a lack of motivation to exercise after admitting only 25% of eligible patients to the study [20]. Another significant reason for the low participation rate was “Too many competing demands on time and energy” [24]. Conversely, Lahart et al. reported a good recruitment rate of 53%, which accepted 80 participants [21]. Only three studies did not provide information about participant recruitment. In other studies, recruitment strategies were reported (email recruitment [18]; screened with the Physical Activity Readiness questionnaire and/or physician [19]; via pathology databases and/or direct referral from the participating hospitals [20]; using an information booklet and contact by the primary researcher via telephone [21]; hospital registry and the attending doctor [24]; multiple method used informational mailings, in-person recruitment, media, and community presentations [26]).

### 3.3. Participants

Participants in the included studies were survivors of various cancers. Four studies included only participants who survived breast cancer [18,21,22,25]. The other two studies involved participants who survived prostate cancer [19,23]. The remaining studies included participants who survived glioma [20], colon, and rectal cancer [26], and the most recent study included a combination of participants after breast cancer and colon cancer [24]. Participants ranged in age from 37 to 78 years. Most studies covered a wide range of age groups.

### 3.4. Control Groups

Instructions for control group participants were reported in seven of nine randomized controlled trials. In six studies, control participants were informed to maintain their usual level of physical activity for daily life throughout the study [18,20,21,22,23,25]. Another study included an active control group under the supervision of a certified personal trainer [19] or only symptomatic supervision without further recommendations [26]. Three of the nine studies offered the control group a variation on the post-study exercise intervention [20,21,26].

### 3.5. Interventions

HB exercise interventions varied across the nine included studies. The intervention period ranged from 12 to 96 weeks. All studies included different forms of remote monitoring and counselling of HB exercise interventions. However, comparing exercises is complicated due to the variety of methods used to assess intensity.

No study involved the real-time monitoring of the exercise session. Exercise protocols included remote post-exercise monitoring and counselling that were performed using a telephone call [19,23,25,26], or email correspondence [20], or a combination of telephone calls and emails [18,21]. Two studies addressed the delivery of monitoring and teleconsulting through regular in-person hospital visits [22,24]. In the studies, monitoring and teleconsulting were mostly performed by exercise physiologists [18,19,23], physiotherapists [20], nurses [24], or a research team member [21,22,25,26]. The consultations ranged from once a week [19,20,23,26] to once every four weeks [21,22], which included monitoring of compliance with exercise intervention (mostly via training diaries), feedback, motivation in the follow-up period, screening barriers to exercise attendance, or adverse events.

The frequency, modality, intensity, duration, and prescribing the exercise interventions varied across studies. In most studies, a range of 3–5 exercise sessions per week was reported (progressive increase in exercise frequency) [19,20,21,22,23,24,26]. One study reported a variable plan of accumulating physical activity (PA; 150 min at medium- to high-intensity PA and 300 min at low to moderate-intensity PA per week) [18]. Six studies included aerobic exercise [18,20,21,22,24,26], two studies included a combination of aerobic and strength training [23,25], and one study included multicomponent exercise [19]. In most studies, the prescription of exercise modality was walking, cycling, or at-home exercise on ergometers [21,23,26]. Two studies included a range of variable aerobic methods, including walking, cycling, ball games, and swimming [20,22]; and one study was walking only [25]. In three studies, the modality of exercise was not adequately defined in the methodological description [18,19,24]. Studies involving resistance training prescribed either bodyweight exercise [19] or resistance bands [23,25]. In one study, flexibility exercise was included as a part of aerobic and resistance training [19]. Warm-up and cool-down phases were included in exercise interventions in three of the nine studies [22,23,26]. Comparing exercise intensity was limited due to the variety of methods used to assess the intensity. In six studies, the exercise intensity was based on prescriptions assessed by calculation and/or estimation of the maximum heart rate (HR) [18,26], or percentage of maximal HR [19,20], or percentage of maximal VO_2_ [22], or first ventilatory threshold [23] examined during the baseline cardiorespiratory exercise test. All six studies used HR monitors to monitor intensity during HB exercise [18,19,20,22,23,26]. The remaining studies prescribed exercise intensity using the rating of perceived exertion (RPE) on the Borg scale (6–20) [24] or a 1–4 scale (light, moderate, vigorous, and very vigorous, respectively) [25], and one study reported that participants exercised only at moderate intensity [21]. The duration of one exercise session ranged from 20 to 45 min in the studies [21,22,23,24,25,26] and two studies used a variable duration of the target summary of exercise minutes per week (150–300 min /week) [18,19]. In one study, the duration of the exercise session was not specified [20]. In four [21,22,23,26] of the nine studies, the intensity and/or duration of exercise was usually increased gradually over the intervention period, but most of these studies lacked a detailed description of the prescribed exercise progression.

### 3.6. Adherence and Compliance

Adherence to the exercise protocol was reported relatively consistently, with only three of the nine studies not reporting exercise adherence [21,25,26]. Adherence was defined in one of two ways: the percentage of completed prescribed exercises or the number or percentage of participants who completed the prescribed number of sessions per week. In five studies, high adherence to intervention was reported (range 71–88%) [19,20,22,23,24]. One study reported a high level of adherence, but did not define their methodology [18]. Adherence to exercise intensity and duration was assessed according to training diaries [23,24,25,26] or web platform training logs [18,19,20]. Two studies reported moderate to high compliance (range of 75–90%) with the target exercise intensity [19,20]. Another study reported a variable intensity exercise prescription [22] and high to excessive compliance with the target session time at the correct exercise intensity per week [18]. Low adherence (17%) in compliance with the duration of exercise prescription was reported in one study [25].

### 3.7. Safety and Adverse Events

Information about the safety of HB exercise interventions was lacking. Adverse events were reported in only four of nine studies [19,20,23,25]. Primarily musculoskeletal adverse events were reported in two cases [19,25] (Table 2). The other two studies reported no serious adverse events [20,23], except one minor deterioration of pre-existing knee pain associated with osteoarthritis.

### 3.8. Methodological Evaluation and Study Quality Results

A description of the methodological evaluation and study quality is reported in Table 2. For all eligible studies, a total of four key methodological elements were assessed (use of randomization for group allocation, use of the unbiased randomization method, blinding of assessors, and use of intention-to-treat analysis). None of the studies met all four quality assessment criteria in this review. Three of the nine studies met a total of three criteria [19,20,21], three studies met two criteria [23,24,26], and three studies met one of four key methodological criteria [18,22,25].

All the studies used randomization to group participants, but two studies did not provide a specific description of the process or the description was insufficient [18,25]. In another five studies, the randomization procedure was stratified [19,20,21,22,24,26] to balance the experimental and control groups, most often by age [20,24,26], in relation to adjuvant therapy [19,21,24], but also by sex [26], or according to the relative VO_2_ classification [20]. The remaining two studies performed a simple randomization procedure without stratification [22,23]. Only three of all studies further reported an unbiased randomization process [19,21,25] using external computer random assignment [19,21] or using sealed envelopes obtained from the research assistant prior to the first data collection [25]. Five studies analyzed data on an intention-to-treat basis [18,20,21,23,24]. For all studies, information was provided on the flow of participants during the study, including those who withdrew or refused to follow-up. Blinding of assessors for at least one of the primary outcomes was performed in a total of three studies [19,20,26]. In the study by Hvid et al. [22], the authors reported that randomization was blinded to referring physicians and laboratory technicians, but not to assessors.

The results of the study quality and reporting (Table A1 in Appendix A) showed that the overall quality was good, with an average score of 10.1 points (range: 8–12 points). The study quality was evaluated with an average score of 3.4 points (range: 2–5 points) and the study reporting was evaluated with an average of 6.6 points (range: 4–7 points). None of the studies was rated as low quality and only one study was rated as high quality score [19].

### 3.9. Outcomes

#### 3.9.1. CRF

The evaluation of the CRF was one of the main criteria for selecting eligible studies for review. CRF was, therefore, part of all nine studies [18,19,20,21,22,23,24,25,26], with four studies having the CRF as the primary outcome [20,21,23,24]. Oxygen consumption (VO_2_peak or VO_2_max) [18,19,20,21,22,23,26] was used in seven studies to evaluate CRF, of which the VO_2_peak/six minute walk test (6MWT) combination [23] was used once and the VO_2_peak/treadwalk test was used once [26]. One study further evaluated CRF using the steep ramp test and exercise test at 70% of maximum workload [24] and one study used only the 6MWT [25].

In four of the nine studies, improvement in CRF was reported in the HB exercise group [18,20,22,23]. Husebo et al. [25] further reported a significant difference from baseline and six months post-treatment for total sample (home-based exercise and usual care). Significant improvements were not reported in the other four studies [19,21,24,26].

A total of eight studies compared CRF results between HB exercise and usual care groups [18,20,21,22,23,24,25,26]. Five of these studies reported a significant CRF improvement in the HB intervention [18,21,22,23,26], and one study reported that the effect ceased to be significant after six months of follow-up [18]. The remaining three studies found nonsignificant differences between groups [20,23,25]. Cornette et al. [23] found a significant difference in per-protocol analysis (not intention-to-treat). Two studies also compared HB and CB exercise interventions [19,24]. Whereas Alibhai et al. [19] found nonsignificant difference between interventions, van Waart et al. [24] reported improved CRF results for the CB in comparison with the HB exercise intervention.

#### 3.9.2. Physical Activity (PA)

A total of seven studies compared the PA levels [18,20,21,23,24,25,26]. Six studies used International Physical Activity Questionnaire [20,21,23,25], Physical Activity Scale for the Elderly [24], and Seven-day Physical Activity Recall [26] questionnaires for evaluation. One study used an accelerometer for PA evaluation [18].

Five of the seven studies reported a significant improvement in PA levels with HB exercise in the intervention group [18,20,23,26], and one study, in the total study sample [25]. In two studies, these results were not reported [21,24]. Cornette et al. [23] reported that the HB exercise group doubled its total PA, but did not describe statistical significance in this context. McNeil et al. [18] reported, in addition to a significant increase in PA levels (both intervention exercise groups), participants reduced sedentary time in the lower-intensity exercise group.

Two of the seven studies further described a statistically significant increase in PA with HB exercise compared with the usual-care group [18,26]; however, Pinto et al. [26] added that group differences attenuated over time. Lahart et al. [21] stated that the effect of the intervention was likely beneficial (91%) on total and moderate PA levels compared with usual care. Another three studies reported a nonsignificant difference between HB exercise and usual care [20,24,25].

#### 3.9.3. Fatigue and HRQOL

The fatigue outcome was measured in five studies using the Functional Assessment of Cancer Therapy–Fatigue [19,26], Multidimensional Fatigue Inventory [23,24], Fatigue Quality List [24], and Schwartz Cancer Fatigue Scale [25] questionnaires. Reporting fatigue results was insufficient in all the studies. Only one study reported results with statistically nonsignificant difference in the HB group [26]. Three studies reported a nonsignificant difference between the HB exercise and usual-care groups [23,25,26]. An exception is the study by van Waart et al. [24], where a significant reduction in fatigue was found in the HB group, but these results were obtained from a subscale of the questionnaire primarily focused on quality of life. A comparison of average scores at baseline and six months post-treatment further showed a nonsignificant difference, suggesting a return to baseline levels of fatigue [25]. One study [24] reported significantly better physical fatigue results in the CB than HB exercise interventions. Alibhai et al. [19] found nonsignificant important changes in fatigue between interventional groups in HB and CB exercise interventions.

A total of four studies investigated the HRQOL [19,23,24,26]. One study reported significant improvement in physical functioning, less nausea and vomiting, and less pain with HB exercise than in the usual-care group [24]. The other two studies found a nonsignificant difference [23,26]. One study reported a 77.4% probability of inferiority (the Functional Assessment of Cancer Therapy–Prostate) of HB to CB exercise [19].

#### 3.9.4. Other Secondary Outcomes

A total of five studies evaluated body composition and/or anthropometry [18,20,21,22,23]; three studies reported a significant improvement in the HB exercise group. McNeil et al. [18] found a significant decrease in body fat mass in the higher-intensity exercise group, and Gehring et al. [20] reported significant reduction in body mass index (BMI) and body weight. Hvid et al. [22] reported a decrease in weight, BMI, and fat mass due to exercise, as well as a significant difference between HB exercise and usual care. Cornette et al. [23] reported a significant increase in BMI in the HB exercise group.

Three studies further evaluated muscle strength levels [19,23,24]. However, only one study reported a significant improvement in strength (handheld dynamometer elbow flexion and knee extension, and grip strength) in comparing the two interventions (CB and HB exercise) [24].

Two studies evaluated the satisfaction of participants with the exercise program [19,20]. In both studies, high satisfaction was found, with no differences between the groups. A further study identified better outcomes for anxiety and depression (Hospital Anxiety and Depression Scale) in HB exercise; however, the difference was nonsignificant compared with usual care [23].

## 4. Discussion

This systematic review provides an overview of the main effects and an assessment of the methodological quality of HB exercise interventions in cancer patients and survivors. To the best of our knowledge, this is the first review article to focus on alternative approaches to rehabilitation after primary treatment of cancer. Only nine clinical randomized controlled trials [18,19,20,21,22,23,24,25,26] were included in this review, which indicates the newness of this type of work. The diversity of the study participants did not allow for data synthesis; nevertheless, the findings of this systematic review suggest that HB exercise interventions can provide a variety of benefits to cancer survivors during the rehabilitation phase. These include improvements in CRF, strength, level of physical activity, HRQOL, and body composition.

Most authors adequately reported the quality of study results, but most trials had limitations in study methodology. Most often, the methods were incompletely described, with details lacking regarding the process of recruiting participants, the period of treatment, or the process of adherence to exercise prescription. These limitations may reduce the overall evaluation of the quality of the systematic review and the generalization of the conclusions, which could be useful for the further development of HB exercise intervention research in the population of cancer survivors. Methodological limitations were present in most studies. None of the studies met all the criteria for methodological rigor described by Spence et al. (randomization process, unbiased randomization, blinding of assessors, and intention-to-treat analysis]. Only three studies met three out of four criteria [19,20,21], three studies met two criteria [23,24,26], and three met one of these criteria [18,22,25]. After reviewing the assessment of study quality and reporting of exercise, only one study out of the nine achieved an evaluation as high quality [19]. This reflects not only the lack of reported information for each study, but also its novelty. As research in this area expands, the methodological quality of studies is expected to improve.

To improve the quality of research in this field of study, the future design of research plans should emphasize methodological quality. In addition to ensuring the quality of the methodology, another finding was that the prescription of exercise interventions varied across studies, complicating the development of general recommendations specific to patients and cancer survivors for the post-treatment period. However, in all studies, HB exercise was found to be feasible and safe, corresponding to supervised exercise interventions results in population cancer survivors [27,28].

The adherence reported in exercise interventions was high, as reported in CB studies. Study participants can be considered sufficiently motivated to exercise in their own environment, without regarding exercise prescription [29,30].

It was found that higher CRF levels may be related to better improvements in adherence, exercise prescription, and fatigue levels [31]. Therefore, the effect of HB exercise interventions in the context of CRF may be crucial, because aerobic exercise is a highly recommended intervention to improve CRF [32]. Current evidence on the prognostic impact of CRF supports the clinical relevance of developing effective strategies to improve CRF levels in cancer patients and survivors [33,34,35].

The systematic review results in the six studies showed a significant CRF improvement after HB exercise intervention compared with the control group [18,21,22,23,25,26]. Previous supervised exercise studies have confirmed the effectiveness of aerobic interventions in a wide range of cancer patients and survivors [36,37,38]. However, there is currently no consensus on the optimal prescription of exercise intensity. Our review shows that exercise for cancer survivors could be prescribed as aerobic exercise two to five times a week, 20 to 50 min per section at 11 to 14 RPE. The dose of exercise time, frequency of exercise per week, and exercise intensity should be gradually increased optimally after consultation with an exercise specialist.

Evidence shows that CRF efficiency can be significantly improved through interval and intensive exercise training [39]. This prescription demonstrated a superior CRF effect over moderate-intensity continuous training exercise and a further reduction in the severity of adverse events such as nausea, vomiting, pain, and physical fatigue [24,40]. Furthermore, it has been noted that the interval exercise protocol is more effective than moderate-intensity continuous exercise in a population of patients with heart failure, which corresponds to the adverse reactions of chemotherapy-induced cardiotoxicity in cancer survivors [41,42].

Less than half of the studies included in this systematic review treated participants with anthracycline, which can cause cardiotoxicity, further progression of heart failure, and reduced left ventricular ejection fraction [43]. Cardiotoxicity associated with cancer treatment may manifest acutely during chemotherapy or later in remission [44]. Accordingly, the safety and efficacy of HB exercise interventions in a population of patients with reduced left ventricular ejection fraction and heart failure were demonstrated [45].

The safety aspect of HB exercise facilities is important. Because patients rehabilitate without in-person supervision, as is common in traditional CB models of rehabilitation, there are some concerns regarding safety. To date, evidence is lacking of the safety of HB exercises. In this review, adverse events were monitored in four of nine studies [19,20,23,25]. As a result, in total (*n* = 98), no participant suffered any serious complication or death associated with HB exercise intervention. Although the results support the implementation of HB interventions in all groups of cancer patients and survivors, more extensive research is needed in this area to adequately determine the safety risk.

Longer-term maintenance is essential after the intervention phase of rehabilitation; only two studies in the review investigated the long-term effect of HB exercise (one year follow-up) [22,26]. Although it would be reasonable to assume that HB exercise will lead to a longer-term improvement in clinical outcomes, more evidence based on methodologically high quality and sufficiently large studies will be needed for confirmation. However, the promising results from individual studies suggest that HB exercise is effective in maintaining several intervention improvements in a long-term strategy.

HB exercise interventions and discussions on this topic are currently crucial given the current COVID-19 pandemic situation, which is associated with public health prevention worldwide, and there are no clear recommendations on how to proceed with the rehabilitation of people with chronic diseases, such as cancer. It was proven that lockdown restrictions and isolation during the COVID-19 pandemic have led to increased anxiety and unhealthy lifestyles, including reduced physical activity [46].

The use of the effective strategies based on the gold standard is currently difficult when the delivery of rehabilitation is reduced or limited worldwide. Thus, innovative methods of social connectivity can be crucial for maintaining patient motivation. This is an even more significant challenge to alternatives such as HB exercise and/or telerehabilitation, which are relevant to all patients with chronic disease [16,47].

Telerehabilitation is an alternative approach for providing long-distance rehabilitation services through information and telecommunication technologies (PC, smartphone, internet, and videoconferencing) [48]. The telerehabilitation model has been successfully investigated in people with various cardiopulmonary diseases [49]. In this systematic review, the five studies on cancer survivors [18,19,20,22,23] met the telerehabilitation form. Based on this, the use of telerehabilitation platforms may lead to increased attractiveness and usefulness in the field of cancer rehabilitation. However, the acceptability and usefulness of the telemedicine approach may be limited by some factors (Table 3).

For example, smartphone-based HR monitors can report results with various accuracy [50]. These data may be unreliable, which can limit the use of mobile-health applications for telemonitoring purposes. Nevertheless, the validity of HR monitors has provided satisfactory accuracy at moderate and intense levels of exercise [51]. Finally, although telemedicine is still unused in the management of chronic diseases, the COVID-19 pandemic has probably renewed interest in using innovative healthcare strategies [52]. Therefore, the HB cancer telerehabilitation model can fill the gap and provide an appropriate alternative approach in the future.

### 4.1. Limitations

Although a comprehensive literature search was performed, it is possible that some eligible studies were not included. Even though scientific articles were included in the systematic review, there could still be bias from some of the published articles not reporting negative findings. The main limitation of most of the included studies was the high variability of the methods used for exercise intervention and for most of the evaluated results. Therefore, it is necessary to determine basic recommendations for outcomes in exercise-based cancer interventions in the future.

The diversity reported in the study characteristics in terms of age and type of cancer in the population involved, as well as in the prescription of exercises (especially different durations, frequencies, and modalities) may have led to a reduction in the overall quality of generalizing findings.

Finally, another limitation is the potential of the appropriate participants included in the studies not being representative of the general population of cancer survivors. The bias could be caused by the selection process, where mainly young or motivated participants who preferred at-home training were included. In addition, many participants did not have the opportunity to express preference for exercise.

### 4.2. Future Directions

For further research on HB exercise intervention evaluations, we recommend researchers to:define the outcomes and evaluate the effect relevant to the phase of cancer rehabilitation,establish a timeframe to define the phase of cancer rehabilitation,determine relevant measurements of outcomes for different cancer populations, andconduct further pilot studies in understudied cancer populations to ensure the feasibility of interventions and data analysis for future research, e.g., sample size determination.

For the future methodological design of clinical trials with sufficient evidence of feasibility, we recommend using:
specified eligibility criteria (clearly stated and fulfilled eligibility criteria),unbiased randomization with description (a description of the randomization method used to allocate patients into study groups should be provided),allocation concealment,blinding of the assessor (for at least one primary outcome), andintention-to-treat analysis.

For studies reporting HB exercise training interventions for cancer survivors during the rehabilitation phase, future studies are recommended to report:
specifications of the time range between cancer treatment completion and study enrolment,participation rate,limitations on recruitment flow and exercise intervention implementation,adherence to exercise intervention and compliance with the exercise prescription,adverse events (any serious medical events, deaths, and hospitalizations),activity monitoring in control groups,exercise volume and energy expenditure (exercise session and intervention duration, session frequency, and exercise intensity and modality),between-group statistical comparisons, andfindings from all analyses conducted including effect sizes.

## 5. Conclusions

Research evidence is lacking in the field of HB exercise interventions in the population of cancer survivors. However, all studies were limited in terms of methodology and reporting of results. Nevertheless, the evidence in this new area, despite the methodological limitations of studies, suggests that HB exercise interventions are feasible, and may provide physiological and psychological benefits for cancer survivors during the rehabilitation period. A methodologically rigorous design for future research is essential for making progress in this field of study.

## Figures and Tables

**Figure 1 cancers-13-01915-f001:**
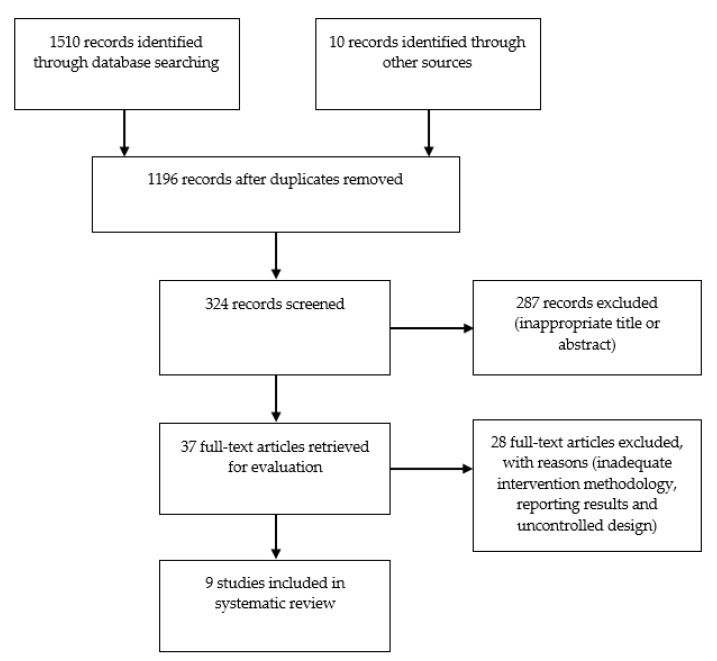
Flow diagram detailing the search strategy.

**Table 1 cancers-13-01915-t001:** Characteristics and results of studies that evaluated the effects of home-based (HB) exercise intervention.

Study	Treatment	Cancer	Study	*n*	Sex	Age	Duration (Week)	Exercise Program	Intensity	Frequency	Monitoring/Feedback	Outcomes Primary/Secondary
McNeil (2019) [18]	Post CT, RT or surgery	BC	RCT (1:1:1)	45	W	58	12	Aerobic exercise; low-intensity group: 300 min/week, high-intensity group: 150 min/week	Low-intensity group: 40–59% HRR; high-intensity group: 60–80% HRR	NS	HR monitor; exercise diary; phone call or e-mail 1/3 weeks	PA/CRF, body composition—anthropometry
Alibhai (2019) [19]	During ADT	PC	RCT (1:1:1)	59	M	70	24	Aerobic exercise + resistance and flexibility exercises; 150 min/week	60–70% HRR, 3–6 RPE (10 on Borg scale)	4–5/week	HR monitor; phone call 1/week	Feasibility/CRF, fatigue, HRQOL, strength, satisfaction, cost-effectivity
Gehring (2018) [20]	Post CT, RT or surgery	Gliom	RCT (2:1)	34	M, W	48	24	Aerobic exercise	60–85% HRmax	3/week	HR monitor; training log online; e-mail 1/week	CRF/PA, body composition—anthropometry, satisfaction
Lahart (2017) [21]	Post ADT or surgery	BC	RCT (1:1)	80	W	52	24	Aerobic exercise; 30 min/session	NS	Gradually 3–7/week	PA diary recommended; phone call 1/4 weeks	CRF/PA, body composition—anthropometry
Hvid (2016) [22]	Post radical prostatectomy	PC	RCT (2:1)	25	M	70	96	Aerobic exercise; 45 min/session	60–65% VO_2_max	3/week	HR monitor; training log; in person control visit 1/4 weeks	NS/CRF, body composition—anthropometry, glucose
Cornette (2016) [23]	During ADT	BC	RCT (1:1)	44	W	51	27	Aerobic exercise, 30–50 min/session + resistance exercise;	1-VT	3/week	HR monitor; exercise diary; phone call 1/week	CRF/PA, fatigue, HRQOL, strength
Van Waart (2015) [24]	During ADT	BC + Colon	RCT (1:1:1)	230	W	51	NS	Aerobic exercise, self-managed PA ≥ 30 min/session	12–14 RPE (6–20 Borg scale)	5/week	Activity diary; in-person control visit 1/CT cycle	CRF, fatigue, strength/PA, HRQOL
Husebo (2014) [25]	During ADT	BC	RCT (1:1)	67	W	52	12	Aerobic exercise, 30 min/session + resistance exercise	At least 2 level of intensity out of 4	7/week	Exercise diary; phone call 1 /2 week	Fatigue, PA/CRF
Pinto (2013) [26]	Postsurgery, CT, or RT	CRC	RCT (1:1)	46	M, W	57	12	Aerobic exercise gradually to 30 min/session	64–76% HRmax	Gradually 2–5/week	Accelerometer; home PA log; 1 phone call/week	PA/CRF, fatigue, HRQOL

CT = chemotherapy, RT = radiotherapy, RCT = randomized controlled trial, HRR = heart rate reserve, HR = heart rate, VO_2_max = maximal oxygen consumption, ADT = androgen deprivation therapy, RPE = Rating of Perceived Exertion, HRQOL = health-related quality of life, HRmax = maximum heart rate, BMI = body mass index, PA = physical activity, PSA = prostate-specific antigen, VT = ventilatory threshold, CRF = cardiorespiratory fitness, NS = not specified, M = men, W = women, N = number of participants, BC = breast cancer, PC = prostate cancer, CRC = colorectal cancer.

**Table 2 cancers-13-01915-t002:** Methodological quality of studies that have evaluated the effects of home-based exercise intervention on cancer patients.

Study	Randomization Process	Unbiased Randomization	Blinding of Assessors	Intention-to-Treat Analysis	Adverse Events	Adherence with Exercise Protocol
McNeil (2019) [18]	NR	NR	NR	Yes	NR	111% of prescribed PA in higher-intensity group (mean 166 min/week), 309% of prescribed PA in lower-intensity group (mean 928 min/week)
Alibhai (2019) [19]	Stratified by duration of prior ADT use	Yes	Yes	No 14 dropouts	Two adverse events grade 2 (primarily musculoskeletal), no events grade 3 or higher	31% (self-reported questionnaire) or 50% (accelerometry) of participants achieved 150 min/week of prescribed PA
Gehring (2018) [20]	Stratified by age, education, tumor grade, disease duration, relative VO_2_ classification, and performance on the letter digit substitution task	NR	Yes	Yes	No adverse events (one aggravation of pre-existing osteoarthritis-related knee pain at 6th month of HB)	Participants completed 79% of prescribed sessions (mean 2.4 sessions/week)
Lahart (2017) [21]	Stratification based on ADT	Yes	No	Yes	NR	NR
Hvid (2016) [22]	Simple adaptive randomization procedure	NR	No	No 6 dropouts	NR	Participants completed 88% of prescribed sessions
Cornette (2016) [23]	Without stratification	NR	NR	Yes	No	Reported only in 14/20 participants, participants completed 88% of prescribed sessions (109% aerobic exercise, 46% resistance exercise)
van Waart (2015) [24]	Stratified by age, primary diagnosis, treating hospital, and ADT use	NR	NR	Yes	NR	55% of participants followed prescribed daily PA levels at least 75% of HB program
Husebo (2014) [25]	NR	Yes	NR	No 7 dropouts	1 knee discomfort, 1 syncope	17% of participants achieved 210 min/week of prescribed PA (58% met the general recommendations of 150 min/week)
Pinto (2013) [26]	Stratified by age, cancer type, and sex	NR	Yes	No 3 dropouts	NR	65% of participants achieved 150 min/week of prescribed PA

NR = not reported, PA = physical activity, ADT = androgen deprivation therapy, VO_2_ = oxygen consumption, HB = home-based exercise.

**Table 3 cancers-13-01915-t003:** Benefits and potential limitations of HB cancer exercise interventions.

Benefits	Limitations
Higher protection from infection	Lack of exercise supervision
Independence in exercise planning	Less face-to-face contact
Less time and/or travel barriers	Lack of social interaction
Integration into daily PA	ICT literacy
Combination with tele-monitoring and/or counselling	Exercise data integration into medical records
Higher privacy	Lack of legal clarity and data protection
Lower costs	Lack of published guidelines

ICT = information and communications technology; PA = physical activity.

## Data Availability

No new data were created or analyzed in this study. Data sharing is not applicable to this article.

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
