# Peer review of "Home-Based Aerobic and Resistance Exercise Interventions in Cancer Patients and Survivors: A Systematic Review"

_cancers, 2021, doi:10.3390/cancers13081915_

Round 1
Reviewer 1 Report
The manuscript was prepared very well. The introduction section justifies the purpose of the study. I congratulate the authors for the preparation of the manuscript
However, I have the following comments:
In the title of the manuscript I would include the type of Exercise interventions: aerobic and resistance... in cancer patients and survivors: A systematic review.
Introduction
- Why have you focused on Home Exercise Interventions? What makes them different from other interventions? clarify this issue
- Is this type of home exercise intervention applicable to what types of cancer? At what stage of disease? Is it compatible with concurrent chemotherapy or radiotherapy treatment? Address this question
Materials and Methods
The methodology is perfectly described and carried out
- Why did the manuscript search begin in 2011?
- “Multimodal interventions were excluded”: What do you want to express?
- In the flowchart 28 studies were excluded, indicate in the flowchart the reason: inadequate results, uncontrolled studies...
Results
- The tables and the text describing them do not require any input, it is the strongest part of this study.
Discussion
- The discussion could be divided into sections: CRF, strength, physical activity level, HRQOL and body composition; for a better understanding.
- Although there is an excellent description of the studies and description of the results, potential mechanisms related to those described in the manuscript could be added. In addition, a brief comparative discussion of the differences between studies could be added.
- Table 3 describes benefits and limitations of exercise: is it the authors' own or is it derived from the articles included? explain this
- a cancer patient profile and an exercise type "standard" should be established at the end of the manuscript.
Author Response
Response to Reviewer 1 Comments
The manuscript was prepared very well. The introduction section justifies the purpose of the study. I congratulate the authors for the preparation of the manuscript
Author response: My co-authors and I would like to thank you for a review of our work. We really appreciate your comments and will address them immediately below.
However, I have the following comments:
In the title of the manuscript I would include the type of Exercise interventions: aerobic and resistance... in cancer patients and survivors: A systematic review.
Author response: Thank you for the above suggestion. The title of the article was modified according to the above recommendation and reads as follows:: "Home-Based Aerobics and Resistance Exercise Interventions in Cancer Patients and Survivors: A Systematic Review"
Introduction
- Why have you focused on Home Exercise Interventions? What makes them different from other interventions? clarify this issue
Author response: Thank you for the above comments. We decided to incorporate home-based exercise interventions because they can provide opportunities/alternatives to supervised rehabilitation programs. Home-based exercise interventions may impact barriers, overall utilization, access to the limitations caused by the COVID-19 pandemic, and/or the use of telemedicine. (The reasons given are described in introduction on page 2, para3) Besides, a comprehensive overview of home-based exercise interventions for cancer survivors is not currently available; therefore, we reviewed the current literature on the role of HB exercise interventions during the rehabilitation period.
- Is this type of home exercise intervention applicable to what types of cancer?
Author response: Thank you for the above suggestion. We believe so. It is indeed the aim of this article to answer whether home-based exercise interventions are feasible. So far, there is no evidence, and the paper describes from another point of view the delivery of the exercises for cancer survivors. The query is addressed in the introduction and discussed in the article on page 2 line 13-16.
(…)”Exercise interventions are increasingly recognized as an important part of treatment and supportive care for patients and cancer survivors [8].”(…)
- At what stage of disease? Is it compatible with concurrent chemotherapy or radiotherapy treatment? Address this question
Author response: Thanks for the comment above. Yes, exercise is compatible with concurrent chemotherapy or radiotherapy. Also, the principle of home-based is based on supervised exercise programs that have been developed since the beginning of the second millennium. The answer is addressed on page XXX. line XXX.
(…)” The scientific evidence from previous systematic reviews suggests that exercise interventions provide a number of physical and psychosocial benefits that can mitigate the effects of cancer treatment.”(…)
Materials and Methods
- The methodology is perfectly described and carried out
Author response: Thank you for the above comment, we appreciate it.
- Why did the manuscript search begin in 2011?
Author response: Thank you for the above comment. The paper provides a summary of studies over the last ten years. The intention was to present studies from the new era after 2010, including new technologies in their use. Besides, until 2010, rehabilitation interventions were supervised programs for population cancers initially and according to the review of Spence et al. 2010 also with low study quality. As with other chronic diseases such as cardiovascular disease, a development in home-based approaches has been reported in this population since 2009 - to date, according to Frederix et al. (2015 A review of telerehabilitation for cardiac patients)
- “Multimodal interventions were excluded”: What do you want to express?
Author response: Thank you for the above query. We have decided that the sentence concerning multimodal interventions is misleading and has been removed from the text.
- In the flowchart 28 studies were excluded, indicate in the flowchart the reason: inadequate results, uncontrolled studies...
Author response: Thank you for the above suggestion. A more detailed indication of the reasons for exclusion has been added according to your recommendation and reads as follows:
|
28 full-text articles excluded, with reasons (inadequate intervention methodology, reporting results and uncontrolled design) |
pg5 fig1
Results
- The tables and the text describing them do not require any input, it is the strongest part of this study.
Author response: Thank you for the above comment, we appreciate it.
Discussion
- The discussion could be divided into sections: CRF, strength, physical activity level, HRQOL and body composition; for a better understanding.
Author response: Thank you for the above suggestion, given that the discussion is conceptualized mainly in the context of the quality of design and reporting and outputs, security, and future directions. We think that proportionally, the distribution as you suggest would not be adequate. These outcomes (CRF, strength, level of physical activity, HRQOL, and body composition) make only about 13% of the discussion text.
- Although there is an excellent description of the studies and description of the results, potential mechanisms related to those described in the manuscript could be added. In addition, a brief comparative discussion of the differences between studies could be added.
Author response: Thank you for the above suggestion; however, it was stated on pg11 line 35 that the studies are difficult to compare in more detail due to the different methodological designs.
- Table 3 describes benefits and limitations of exercise: is it the authors' own or is it derived from the articles included? explain this
Author response: Thank you for the above suggestion. Table No. 3 is the author's own and cannot be formulated to be derived from the articles.
- a cancer patient profile and an exercise type "standard" should be established at the end of the manuscript.
Author response: Thank you for the above suggestion. We have incorporated your above proposal into the discussion text on page 11 and reads as follows: "Our review shows that exercise for cancer survivors could be prescribed as aerobic exercise 2 to 5 times a week, 20 to 50 minutes per section at 11 to 14 RPE. The dose of exercise time, frequency of exercise per week, and exercise intensity should be gradually increased optimally after consultation with an exercise specialist."
Reviewer 2 Report
Dear authors,
Your manuscript is interesting but I need you to answer some questions:
ABSTRACT
- Web of Science is not a database but a metasearch engine.
INTRODUCTION
- The verb "summarize" is not suitable for a systematic review. I suggest that authors use verbs such as: describe, analyze, identify ...
MATERIALS AND METHODS
2.1. Search Strategy:
- Why do you exclude studies with only resistance exercises? Do you also exclude trials with strength exercises? They don't mention it in the criteria. Justify your answer.
RESULTS
- You have few results. I suggest that authors do a manual search in the references of your 9 results.
REFERENCES
- Many bibliographies are obsolete. The bibliographic citations used are more than 5 years old (34.8 % without considering "results"). The authors must update and arrange the bibliography.
Author Response
Response to Reviewer 2 Comments
Dear authors,
Your manuscript is interesting but I need you to answer some questions:
Author response: My co-authors and I would like to thank you for a review of our work. We really appreciate your comments and will address them immediately below.
ABSTRACT
- Web of Science is not a database but a metasearch engine.
Author response: Thank you for the above suggestion. We have incorporated a more precise designation of "metasearch engine" as you requested and reads as follows:
pg.2 (…)” Search strategies and article selection criteria were searched from January 2011 to January 2021 through the PubMed database and the Web of Science metasearch engine “(…)
INTRODUCTION
- The verb "summarize" is not suitable for a systematic review. I suggest that authors use verbs such as: describe, analyze, identify ...
Author response: Thank you for the above suggestions. We agree with your opinion and therefore we have corrected the word "summarize" throughout the text of the systematic review.
pg.1 (…) “This article identifies the literature on the health effects of HB exercise interventions “(…)
pg.1 (…)”The aim of this descriptive systematic review was to identify the literature focusing on the health effects of HB exercise interventions “(…)
pg.2 (…)”systematic review was to identify the literature on the health effects of HB exercise interventions “(…)
MATERIALS AND METHODS
2.1. Search Strategy:
- Why do you exclude studies with only resistance exercises? Do you also exclude trials with strength exercises? They don't mention it in the criteria. Justify your answer.
Author response: Thank you for the above suggestion. We decided not to include interventions based only on strength training, as they not significantly affect cardiorespiratory fitness. This review is based on studies that have evaluated cardiorespiratory fitness. We believe that this is an important indicator/focus in exercise programs in cancer survivors. The same is true in the rehabilitation of chronic diseases such as cardiovascular, where aerobic component and improvement of cardiorespiratory fitness are considered the exercise program's primary goal. It was confirmed that this indicator has an impact on disease prognosis and even mortality (Kavanagh, et al. "Prediction of long-term prognosis in 12 169 men referred for cardiac rehabilitation." Circulation (2002): 666-671.). The "strength only” exercise interventions are listed as part of the exclusion criteria; on the pg 3, line 13-14:
(…)”Interventions that included only resistance exercises were excluded”(…)
RESULTS
- You have few results. I suggest that authors do a manual search in the references of your 9 results.
Author response: Thank you for the above comment. A manual search has been performed and the description is addressed in the text on page 3 line 27-28.
(…)”reference lists of localized articles on the topic were screened.”(…)
REFERENCES
- Many bibliographies are obsolete. The bibliographic citations used are more than 5 years old (34.8 % without considering "results"). The authors must update and arrange the bibliography.
Author response: Thank you for the above query. There was probably a miscalculation on your side.
In our evaluation, the bibliographic citation states the age:
Total n=52
Exclude n=9 (results)
Total n=43 to review:
2016 or new = n=25 (more than half of the references are 5 years old!)
18 of 43 references are old (which means only 43%)
Of the complete list (n = 52) is 60% no more 5 years old, which we think is appropriate, and we would even say that such a portion is above standard. In our opinion, it can be stated that the article identifies and discusses the latest research in its content. (33 percent from the whole article are references 2 years old)
In addition to the above, after a detailed reference review, we incorporated two new references, "under 5 years":
42 Freyssin, C.; Verkindt, C. Prieur, F.; Benaich, P.; Maunier, S.; Blanc, P. Cardiac rehabilitation in chronic heart failure: effect of an 8-week, high-intensity interval training versus continuous training. Arch. Phys. Med. Rehabil. 2012, 93, 1359-1364, doi:10.1016/j.apmr.2012.03.007.
42 Papathanasiou, JV.; Petrov, I.; Tokmakova, M.P.; Dimitrova, D.D.; Spasov, L.; Dzhafer, N.S.; Tsekoura, D.; Dionyssiotis, Y.; Ferreira, A.S.; Lopes, A.J. et al. Group-based cardiac rehabilitation interventions. A challenge for physical and rehabilitation medicine physicians: a randomized controlled trial. Eur. J. Phys. Rehabil. Med. 2020, 56, 479-488, doi:10.23736/S1973-9087.20.06013-X.
45 O'Connor, C.M.; Whellan, D.J.; Lee, K.L.; Keteyian, S.J.; Cooper, L.S.; Ellis, S.J.; Leifer, E.S.; Kraus, W.E.; Kitzman, D.W.; Blumenthal, J.A. et al. Efficacy and safety of exercise training in patients with chronic heart failure: HF-ACTION randomized controlled trial. JAMA. 2009, 301,1439-1450, doi:10.1001/jama.2009.454.
45 Gielen, S.; Laughlin, M.H.; O'Conner, C.; Duncker, D.J. Exercise training in patients with heart disease: review of beneficial effects and clinical recommendations. Prog. Cardiovasc. Dis. 2015, 57: 347-355, doi:10.1016/j.pcad.2014.10.001.
Round 2
Reviewer 2 Report
Dear authors,
Thanks for your reply. The explanations of the authors are satisfactory. The paper has greatly improved its quality.
Congratulations on your work.
Best regards